# Surface Reformation of Medical Devices with DLC Coating

**DOI:** 10.3390/ma14020376

**Published:** 2021-01-14

**Authors:** Mao Kaneko, Masanori Hiratsuka, Ali Alanazi, Hideki Nakamori, Kazushige Namiki, Kenji Hirakuri

**Affiliations:** 1Department of Electrical and Electronic Engineering, Tokyo Denki University, Tokyo 120-8551, Japan; hirakuri@mail.dendai.ac.jp; 2Nanotec Co., Ltd., Chiba 277-0872, Japan; hiratsuka@nanotec-jp.com (M.H.); nakamori@nanotec-jp.com (H.N.); 3Department of Biomedical Engineering, King Saud University, Riyadh 11362, Saudi Arabia; asanazi@KSU.EDU.SA; 4Namiki-Mi Co., Ltd., Tokyo 132-0035, Japan; namiki@namiki-mi.co.jp

**Keywords:** intracorporeal medical device, DLC coating, ionized deposition system, surface treatment, low friction, durability, bodily acid, sterilization, anti-reflection, adhesion

## Abstract

We evaluated the adhesion, friction characteristics, durability against bodily acids, sterilization, cleaning, and anti-reflection performance of diamond-like carbon (DLC) coatings formed as a surface treatment of intracorporeal medical devices. The major coefficients of friction during intubation in a living body in all environments were lower with DLC coatings than with black chrome plating. DLC demonstrated an adhesion of approximately 24 N, which is eight times stronger than that of black chrome plating. DLC-coated samples also showed significant stability without being damaged during acid immersion and high-pressure steam sterilization, as suggested by the results of durability tests. In addition, the coatings remained unpeeled in a usage environment, and there was no change in the anti-reflection performance of the DLC coatings. In summary, DLC coatings are useful for improving intracorporeal device surfaces and extending the lives of medical devices.

## 1. Introduction

In recent years, owing to the continuous development of medical technology, the average length of life has increased significantly in developed countries [1]. As a result, the number of deaths caused by illnesses related to aging is continuously increasing, and treatments for early detection of illnesses and less invasive medical procedures are attracting attention [2]. The early detection of cancer is of particular importance because it significantly reduces the mortality rate [3]. However, medical devices have material-related problems. Speculum and cervical dilators that are widely used for the examination and treatment of cervical and vaginal cancers are made from stainless steel (SUS) and brass [4]. When performing examinations and surgery, such as a speculum diagnosis, the reflection of surgical light or a laser scalpel on metal surfaces is a cause of interference in treatment and examination [5]. Therefore, current intracorporeal medical devices, such as speculums, need to be coated with a matte black coating, such as black chrome plating, that suppresses light reflection [4]. However, metal materials, such as black chrome plating, have a high coefficient of friction on the contact surface with the living body, causing discomfort and pain stemming from the friction experienced by the patient [6]. In addition, depending on the part of the biologic organ into which a device is inserted, the metal material may elute because of acid in the body, thereby causing problems like the deterioration of the black chrome plating and skin inflammation in patients with metal allergies [7]. It is therefore highly important to treat medical devices after use to prevent infectious diseases, and, once used, medical devices must be sterilized in actual clinical settings [8]. The deterioration of coating materials during the medical processes is also an issue [9]. For these reasons, there is a demand for the development of surface treatment technology for medical devices as an alternative to black chrome plating.

Diamond-like carbon (DLC) is a non-crystalline or amorphous carbon-based material (lacking a crystalline structure) consisting of sp^2^-bonded carbon atoms of a graphite structure and sp^3^-bonded carbon atoms of a diamond structure [10]. Some coatings contain hydrogen at 0–40 atm%, and their properties depend on the ratio between sp^2^ bonding and sp^3^ bonding with the hydrogen content [11]. Therefore, DLC coatings are produced under conditions suitable for intended use and the properties on the material surfaces. By reducing the hydrogen content, a coating that is nearly as hard as diamond will be formed, whereas in the presence of hydrogen, a coating with gas barrier properties and chemical resistance will be formed [12]. Owing to its high hardness, gas-barrier properties, low-friction properties, and other features depending on the coating formation methods, DLC coating continues to be developed in industrial applications, such as surface coating of cutting tools and optical parts [13]. In addition, because DLC has properties like chemical stability and biocompatibility, it is attracting attention as a surface modification for medical devices with DLC coated on biomaterials [14,15,16,17,18].

This study is intended to utilize DLC surface coating technology to impart functions such as sliding properties, chemical resistance, sterilization resistance, and surgical light reflection prevention in various intracorporeal medical devices. Therefore, in this experiment, film adhesion was evaluated using a scratch test as a mechanical property of DLC coatings. To evaluate the sliding property within a living body under conditions simulating clinical settings, the coefficient of friction was measured in a simulated body environment using lubricating liquids. To evaluate durability, an immersion test simulating the acid in the body and a high-pressure steam sterilization test were conducted repeatedly. Furthermore, regarding the reflection characteristics for surgical light, a sponge-rubbing test simulating the usage environment was performed.

## 2. Materials and Methods 

### 2.1. Sample Preparation and Surface Treatment

In this experiment, we used SUS304 samples (Namiki-Mi Co., Ltd., Tokyo, Japan), which have a low cost and high workability, and are widely used as actual medical devices [19]. In the scratch test, durability test, and evaluation of light reflection characteristics, SUS304 (20 mm × 20 mm × 1.5 mm) samples were used as analysis samples. In the simulated body friction coefficient test, SUS304 (40 mm × 100 mm × 1.5 mm) samples were used to match the size with the simulated living body tester.

Black chrome plating was applied using electroplating [20]. Film formation was performed with SUS used as a cathode and the plating material (Cr) used as an anode. A schematic diagram of the electroplating method is shown in Figure 1. The sample on which black chrome was formed on SUS304 is denoted as Black Cr/SUS. In the Black Cr/SUS samples, film formation with a thickness of 1.8 µm was confirmed by cross-sectional scanning electron microscopy (SEM) (JEOL, Tokyo, Japan). 

DLC was formed via the ionized evaporation method using C_6_H_6_ as a source gas [4,21]. DLC was deposited under the conditions of base pressure 2 × 10^−3^ (Pa), gas pressure 0.2 (Pa), filament current 30 (A), substrate voltage 2 (kV), temperature 200–250 (°C), and deposition time 220 (min). The samples for film formation were subjected to ultrasonic cleaning in an acetone bath as a pretreatment [10]. The sample in which DLC was formed on SUS304 is denoted as DLC/SUS. In the DLC/SUS sample, film formation with a thickness of 1.5 µm was confirmed by cross-sectional SEM. Friction strongly depends on the surface condition (roughness and electrical potential) and the usage environment (dry or wet) of the material [4,22,23,24,25]. DLC coatings are water repellent and, as such, cannot be expected to effectively reduce the coefficient of friction when using lubricating liquids [26]. Therefore, to measure the coefficient of friction in a simulated body environment, the DLC/SUS samples were subjected to oxygen plasma surface treatment using the high-frequency plasma method to improve the hydrophilicity of the outermost surface [27]. The oxygen plasma surface treatment reformed the DLC/SUS surface with plasmaized oxygen. The processing conditions were pressure 10 (Pa), power supply 200 (W), and treatment time 2 (min). The DLC/SUS sample subjected to oxygen plasma treatment was denoted as O-DLC/SUS. A schematic of the plasma reactor and the surface treatment conditions have also been shown in previous research [26].

### 2.2. Evaluation of Film Adhesion by Scratch Testing

To ensure reliability, thin-film coatings must not peel off in a usage environment when used as a surface treatment for medical devices. Films with weak adhesion will be peeled off, and these films will not be able to exhibit their functions in the peeled area. In this study, the adhesion was evaluated for the black chrome plating and the DLC-coated SUS sample using a scratch tester (Revetest Scratch Tester) (CSM Instruments, Switzerland). The conditions were set with reference to the ISO 20502 standard in the experiment [28]. In the scratch test, Black Cr/SUS and DLC/SUS samples were measured three times each, and the measured values were averaged to determine the peeling load. A schematic of the scratch test is shown in Figure 2, and the conditions for the scratch test are shown in Table 1.

### 2.3. Measurement of the Coefficient of Friction in a Simulated Body Environment

To clarify the effect of friction between the Black Cr/SUS and DLC/SUS samples and biomaterials, a simulated body environment was prototyped using urethane to measure the friction coefficient [26]. To measure the coefficient of friction in a simulated body environment, the DLC/SUS samples were subjected to oxygen plasma treatment using the high-frequency plasma method and then surface-treated. To calculate the coefficient of static friction, both sides of the sample were deposited in DLC and were treated with oxygen plasma. In this test, a comparative study of two samples, Black Cr/SUS and O-DLC/SUS, was conducted. In addition to the measurement under dry conditions, measurements were also performed under wet conditions containing blood, physiologic saline, and hyaluronic acid solution to simulate an actual clinical setting. Blood (Tokyo Shibaura Zoki Ltd., Cow blood, Tokyo, Japan), physiologic saline (SIGMA, sodium chloride solution 0.9%), and hyaluronic acid solution (FUJIFILM, hyaluronic acid sodium 0.1%) were used as lubricating liquids. These lubricating liquids were selected because they have high biocompatibility and are commonly used in clinical tests [29].

The specific test procedures for measuring the coefficient of friction in a simulated body environment were the same as those used in previous research [26].

### 2.4. Immersion and Sterilization Tests with Hydrochloric Acid

Immersion and sterilization tests with hydrochloric acid were conducted to study the acid resistance and durability of black chrome coatings and DLC coatings in a simulated clinical setting.

The Black Cr/SUS and DLC/SUS samples were used in these tests. A schematic of a high-pressure steam sterilizer and the sterilization conditions have been shown in previous research [26]. A schematic of the immersion test with hydrochloric acid and the immersion conditions have also been shown in previous research [26].

The chemical composition of DLC/SUS was analyzed by using X-ray photoelectron spectroscopy (XPS) (JEOL, Tokyo, Japan). The surface conditions of the samples were observed with a Schottky field-emission scanning electron microscope (FE-SEM: JEOL, JSM-7100F) (JEOL, Tokyo, Japan). In addition, surface elements were measured with an energy-dispersive X-ray spectrometer (EDS; JEOL, JED-2300) (JEOL, Tokyo, Japan). The results of these tests using Black Cr/SUS and DLC/SUS samples were compared to investigate the utility of the DLC coatings.

### 2.5. Light Reflection Evaluation before and after the Cleaning Test

A black coating is applied to medical devices used in tests and treatments to prevent the reflection of surgical light [6]. However, depending on the cleaning and usage environment, the coating material may peel off and subsequently become unable to prevent the reflection of surgical light. Therefore, to study the surgical light reflection characteristics before and after the use of medical devices, a cleaning test with a sponge (ICHIGUCHI Ltd., outer diameter: 50 mm) (ICHICUCHI ltd., Yokohama, Japan) was conducted on each sample before and after use. The surgical light reflection characteristics were evaluated via specular gloss measurement using a gloss meter (HORIBA, IG-331) (HORIBA, Kyoto, Japan). The SUS, Black Cr/SUS, and DLC/SUS samples were used in the test. Using the glossiness of SUS as a reference, the glossiness of Black Cr/SUS and DLC/SUS was measured before and after use. The specific test procedures are shown below. A schematic of the test is shown in Figure 3. The test conditions are listed in Table 2.

(1)Each substrate was fixed to a drill press (MECANIX Co., Ltd. Shop-Ace M18A) (MECANIX Co., Ltd., Japan).(2)The sponge attached to the drill press was slid at a constant load.(3)The reflection characteristics of each sample after sliding were measured using a gloss meter.

After sliding, the surface condition of the sample on each substrate was visually observed. The results of these tests using the Black Cr/SUS and DLC/SUS samples were compared to investigate the utility of the DLC coatings.

## 3. Results and Discussion

### 3.1. Film Adhesion Evaluation by Scratch Testing

As previously mentioned, to ensure reliability, thin-film coatings must not peel off in a usage environment when used as a surface treatment for medical devices. A scratch test was conducted on the Black Cr/SUS and DLC/SUS samples to measure the film adhesion. Table 3 shows the delamination loads measured in the scratch test. Each sample was measured three times to calculate the average. The results in Table 3 show that the delamination loads of Black Cr/SUS and DLC/SUS are 3.04 and 24.01 (N), respectively. The DLC coatings demonstrated a load withstanding capacity that was approximately eight times larger than that of black chrome plating. Delamination loads of DLC coating was roughly the same value as other research paper [30]. In addition, delamination loads of DLC coating was higher than other materials [31]. These results suggest that when DLC coatings are formed on medical devices, their mechanical adhesion rate and usage life are higher than those of the substrate coated with black chrome plating. 

### 3.2. Simulated Body Coefficient of Friction Measurement in Each Lubricating Liquid Environment

To clarify the effect of friction between the Black Cr/SUS and DLC/SUS samples and biomaterials, a simulated body environment was prototyped using urethane to measure the friction coefficient. Simulated body friction coefficient testing was conducted using blood, physiologic saline, and a hyaluronic acid solution, as well as under dry conditions. The measurement results are shown in Figure 4. The friction coefficient in this study was measured by sandwiching the substrate with urethane rubber. Therefore, the friction coefficient in this study is higher than the normal friction coefficient. The coefficient of friction under dry conditions was 1.391 for Black Cr/SUS and 0.763 for O-DLC/SUS. In addition to its lower friction in dry conditions, O-DLC/SUS showed lower frictional properties than Black Cr/SUS under any lubricant conditions. In addition, the coefficient of friction of O-DLC/SUS under physiological saline and hyaluronic acid conditions was lower than that under dry conditions. These results suggest that O-DLC coatings have a lower friction within the living body than black chrome plating, and it is highly possible that physiological saline and hyaluronic acid can suppress the invasiveness of certain medical procedures caused by friction. There is no specific value for the optimum friction coefficient in the body. For a medical device to be inserted into the body, the lower the friction coefficient, the better.

### 3.3. Immersion and High-Pressure Steam Sterilization Tests with Hydrochloric Acid

To study the acid resistance and durability of Black Cr/SUS and DLC/SUS in a simulated clinical setting, immersion and sterilization tests with hydrochloric acid were performed to evaluate the durability before and after the immersion and sterilization tests. Chemical composition was analyzed by using X-ray photoelectron spectroscopy (XPS). The surface state and film composition were measured using field-emission scanning electron microscopy (FE-SEM) and energy-dispersive X-ray spectroscopy (EDS). The C1s spectra of DLC/SUS before and after the immersion and sterilization tests are shown in Figure 5. These spectrum were deconvoluted to find the ratio of sp^3^/(sp^2^ + sp^3^). Binding energies corresponding to both the diamond and graphite were found at 284.3 eV (sp^2^) and 285.0 eV (sp^3^) before and after the immersion and sterilization tests. The sp^3^/(sp^3^ + sp^2^) ratio governing the quality of the DLC films was calculated from the C1s peak (a greater quantity of sp^3^ bonds makes the films more similar to diamond-like film, while more sp^2^ bonds make them more graphite-like). The sp^3^/(sp^2^ + sp^3^) before and after the immersion and sterilization tests showed 0.23 to 0.26, and there was almost no change before and after testing. The results of XPS confirmed that the surface of DLC/SUS was stable before and after immersion and sterilization.

SEM images before and after the tests are shown in Figure 6. The figure demonstrates that the surface of Black Cr/SUS after the immersion and sterilization tests was changed and damaged by acid and sterilization. On the other hand, no surface deterioration was confirmed with the DLC/SUS sample before and after the immersion and sterilization tests. This is presumably ascribable to the chemical stability, which is a characteristic of DLC coatings. 

To analyze significant morphological changes in the Black Cr/SUS surface before and after the immersion and sterilization tests, a quantitative analysis of elements was performed using EDS. In addition to Fe, Cr, and Ni, which are the main elements of SUS, as well as C, which is the element of DLC, studies were mainly conducted on O to analyze the possibility of oxidation. The analysis results are shown in Table 4. For the Black Cr/SUS sample, the proportion of Cr decreased from 63.85 to 50.25, whereas the proportion of O increased after the immersion and sterilization tests. This is because the passive film was dissolved by the hydrochloric acid used in the immersion test. Moreover, the moisture from the sterilization process invaded the boundary between the material and the film, causing the Black Cr/SUS sample to corrode [32,33]. Next, we considered the DLC/SUS sample. Neither an exposure of metal material nor a change was confirmed with the DLC/SUS sample before and after the immersion and sterilization tests. In other words, the surface of DLC/SUS was stable before and after the immersion and sterilization tests, and its durability as a coating material was confirmed. The results of SEM and EDS suggest that DLC coatings are more durable than conventional coatings as a surface treatment technology for medical devices.

### 3.4. Light Reflection Characteristic Evaluation before and after the Cleaning Test

To study the surgical light reflection characteristics before and after the use of medical devices, a cleaning test with a cleaning sponge was conducted on each substrate before and after use. The surgical light reflection characteristics were evaluated with a gloss meter. The measurement results of the glossiness are shown in Table 5, and the surface conditions before and after sliding are shown in Figure 7. The glossiness of SUS was 75.4, which was used as the reference. The glossiness of the Black Cr/SUS sample was 6.8 before sliding and 63.8 after sliding. It is plausible that the function as a light reflection inhibitor deteriorated because we confirmed that the black chrome plating was peeled off and that the SUS was exposed by cleaning. The coating material with the lowest glossiness indicated approximately 0 to 30 [34]. On the other hand, the glossiness of the DLC/SUS sample before and after sliding was 15.8 and 15.4, respectively. No change was observed in the reflection characteristics before and after sliding, thereby suggesting that the DLC coatings are useful as a light reflection inhibitor in a usage environment.

## 4. Summary and Conclusions

In this study, we evaluated the adhesion, friction characteristics, durability against bodily acids, sterilization and cleaning, and light reflection resistance of diamond-like carbon (DLC) formed as a surface treatment for intracorporeal devices. The coefficient of friction was lower with DLC coatings than with black chrome plating in all environments. Furthermore, DLC showed an adhesion of approximately 24 N, which is eight times stronger than that of black chrome plating. The DLC samples were also significantly stable without being damaged during immersion in acid and high-pressure steam sterilization, which were conducted as a durability test. In addition, the coatings remained unpeeled in a usage environment, and there was no change in the anti-reflection performance of the DLC coatings. In summary, the use of DLC coatings is a useful technique for improving the surfaces of intracorporeal devices and extending the life of medical devices.

## Figures and Tables

**Figure 1 materials-14-00376-f001:**
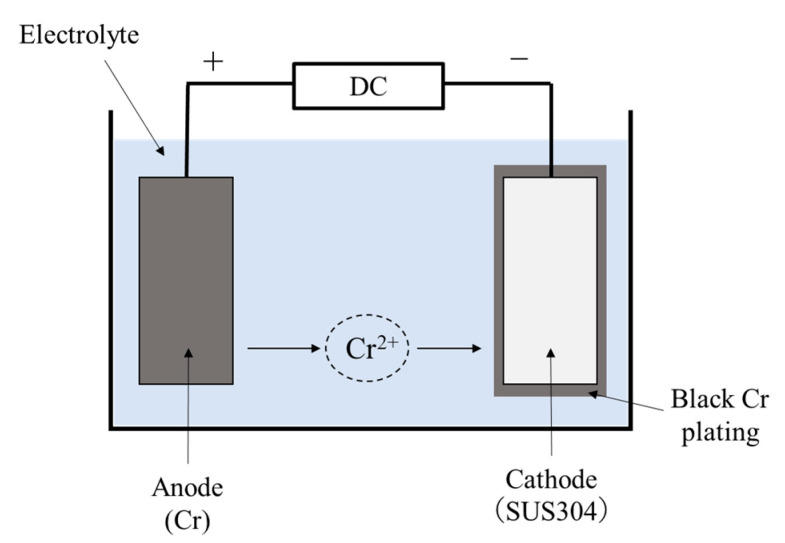
Schematic diagram of the black chrome (Black Cr) plating.

**Figure 2 materials-14-00376-f002:**
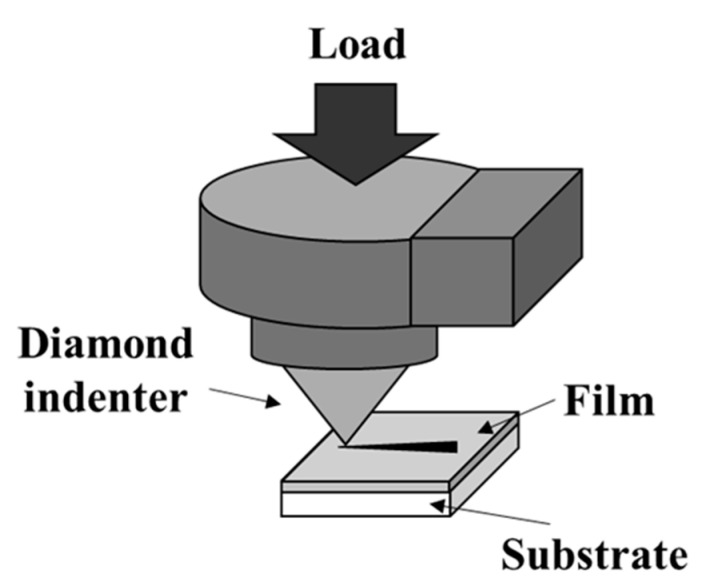
Schematic of the scratch test to measure adhesion.

**Figure 3 materials-14-00376-f003:**
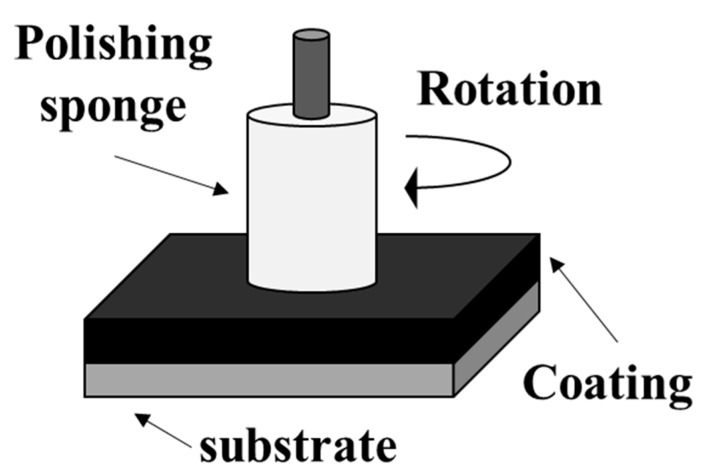
Schematic of the cleaning test.

**Figure 4 materials-14-00376-f004:**
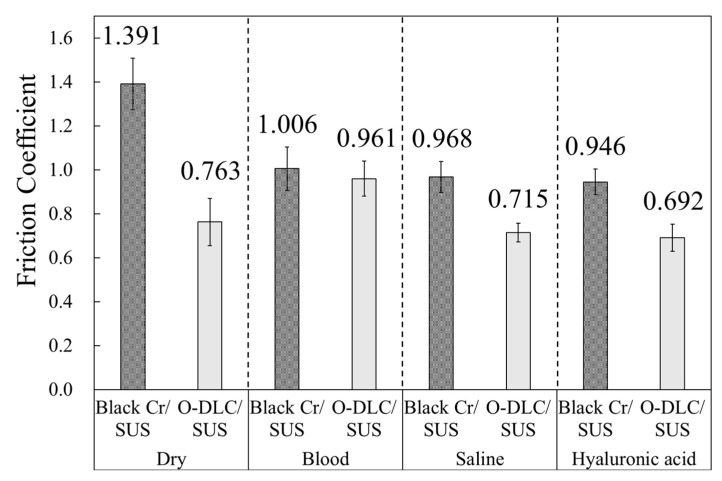
Coefficient of friction of each sample in a simulated body environment.

**Figure 5 materials-14-00376-f005:**
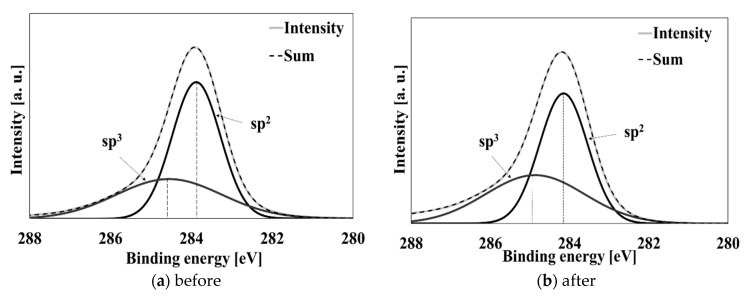
X-ray photoelectron spectroscopy (XPS) spectra for diamond-like coating (DLC)/SUS (**a**) before and (**b**) after the immersion and sterilization tests.

**Figure 6 materials-14-00376-f006:**
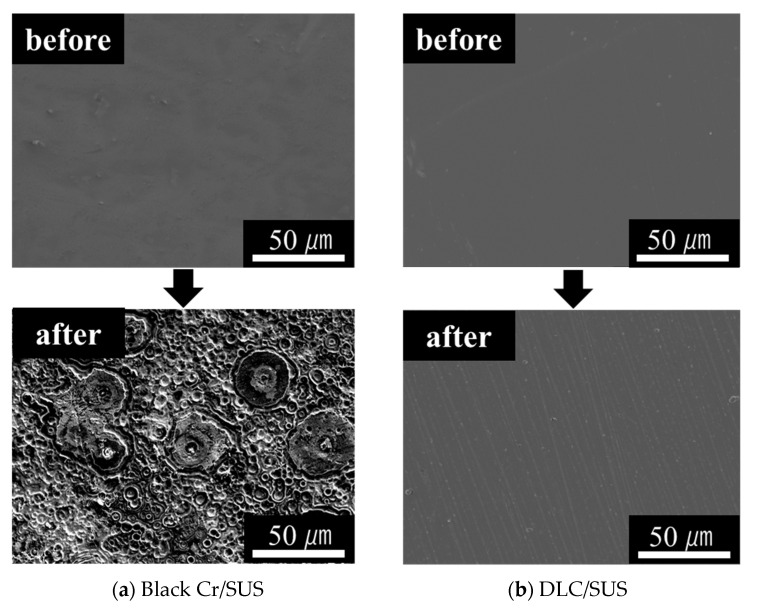
Scanning electron microscopy (SEM) images of (**a**) Black Cr/SUS; (**b**) DLC/SUS before and after immersion and sterilization tests.

**Figure 7 materials-14-00376-f007:**
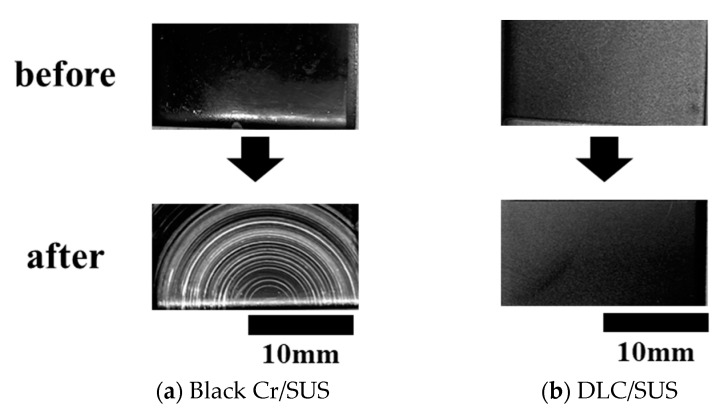
Surface image of (**a**) Black Cr/SUS and (**b**) DLC/SUS after cleaning test.

**Table 1 materials-14-00376-t001:** Scratch test conditions.

Samples	Black Cr/SUS, DLC/SUS
Scratch speed (mm/min)	10
Applied load increase rate (N/min)	100
Initial load (N)	0.9
End load (N)	60
Temperature (°C)	24
Humidity (%)	48

**Table 2 materials-14-00376-t002:** Conditions of the cleaning test.

Substrate	Black Cr/SUS, DLC/SUS
Diameter of polishing sponge (mm)	50
Load (N)	10
Rotation speed (rpm)	1290
Time (s)	20

**Table 3 materials-14-00376-t003:** Delamination loads of each sample measured in the scratch test.

Sample	Delamination Load (N)				
	N1	N2	N3	Average	Standard Deviation
Black Cr/SUS	2.85	3.63	2.64	3.04	0.43
DLC/SUS	21.81	23.98	26.25	24.01	1.81

**Table 4 materials-14-00376-t004:** Surface element analysis of samples using energy-dispersive spectroscopy (EDS).

wt.%	Black Cr/SUS		DLC/SUS	
	Before	After	Before	After
C	2.5 ± 0.21	2.3 ± 0.56	98.1 ± 0.31	98.7 ± 0.28
O	32.2 ± 0.47	47.2 ± 0.43	1.9 ± 0.12	1.3 ± 0.34
Fe	1.2 ± 0.46	0.3 ± 0.20	ND	ND
Cr	63.9 ± 0.38	50.3 ± 0.64	ND	ND
Ni	0.2 ± 0.27	ND	ND	ND

Mean ± SD, Not Detected (ND): < 0.02%.

**Table 5 materials-14-00376-t005:** Glossiness of samples before and after the cleaning test.

Sample	Glossiness (%)	
	before	after
SUS	75.4 (standard)	
Black Cr/SUS	6.8	63.8
DLC/SUS	15.8	15.4

## Data Availability

The date presented in this study are available on request from the corresponding author.

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
