# Peer review of "Surface Reformation of Medical Devices with DLC Coating"

_materials, 2021, doi:10.3390/ma14020376_

Round 1

Reviewer 1 Report

Kaneko and coworkers have tribological, adhesion, and surface properties of DLC for biomedical applications. The authors have evaluated DLC under realistic clinical conditions, which is interesting for medical applications. Hence, DLC seems to be a good alternative to black chrome. This is an interesting work, but there are many issues to be addressed prior to acceptance of this manuscript.

1) Please define what SUS304 stands for.

2) Please provide more details on electroplating. It must be possible to reproduce your experiments by other researchers.

3) Please provide base pressure, temperature, and geometry used to deposit DLC.

4) Please provide more details on O plasma treatment of DLC.

5) The authors have not addressed the chemical composition of DLC adequately. What is the H content? What about other impurities in DLC? Furthermore, what is the ratio of sp2 to sp3 bonding? This is essential for performance of DLC. What happens with the surface composition of DLC under exposure to various chemicals (e.g. hydrochloric acid). This is the major shortcoming of the manuscript.

6) The authors have ignored many theoretical papers on DLC (e.g. Phys. Rev. B 48, 4823 (1993); Phys. Rev. B 54, 9703 (1996); Tribol. Int. 77, 15 (2014); etc.). Please discuss your results with respect to the theoretical insights on DLC.

7) Generally, a more detailed comparison with literature is missing. For instance, the delamination loads should be compared with literature on DLC and other materials.

8) Why is the friction coefficient of DLC so high? Please discuss it and compare with literature on DLC and other materials.

9) Magnification in Fig. 4 may be too small. Please elaborate/improve.

10) EDS is not as precise as claimed in Table 5. Please decrease the number of digits and provide error bars.

11) Please compare the obtained light reflection data on DLC with literature.

12) English is satisfactory, but there are grammatical errors (e.g. “…is used traditional…” in abstract). Some statements make no sense (e.g. “…DLC coatings are a useful technique…” in the abstract – coatings are not techniques). Please proof read the whole manuscript and improve it.

Reviewer 2 Report

The authors analyzed the adhesion, friction characteristics of diamond-like carbon and explained it in depth through well-organized characterization methods. Overall, This work is interesting. I suggest this paper should be accepted after minor revisions. However, there are still some issues need to be solved. Some detailed comments are as follows:

P2, Line 77 “A SUS304 sample upon which black chrome was coated was represented as Black Cr/SUS.” I can't understand this sentence.

P2, Line 78 How does the author determine that the thickness of the film is 1.8mm?

P3, Line 98 “In this study, the adhesion was evaluated for the black chrome plating film and the DLC-coated SUS sample using a scratch tester” instead of “In this study, the adhesion was evaluated between the black chrome plating film and the DLC-coated SUS sample using a scratch tester”

P3, Line 100 “The conditions were set with reference to the standard ISO 20502 in the experiment” instead of “In this measurement, the conditions were set with reference to the ISO 20502 standard”

P5, Line 154, Author uses the Eq. (2) to calculate the coefficient of static friction, are both sides of the sample deposited the DLC coatings?

P6, Line 213, why does the author chose 20 seconds as the cleaning test time? Actually, I personally think the cleaning test time is brief.

P7, Line 228, “To clarify the effect of friction between Black Cr/SUS and DLC/SUS samples and biomaterials” I can't understand this sentence.

P7, Line 232, DLC coatings can reduce the coefficient of friction (the smaller the better), but is there an optimal coefficient of friction for body environment?

P8, Line 279, For the dimensioning, spaces are required between numbers and letters in Fig.4. e.g. “50 μm” instead of “50μm”

Reviewer 3 Report

The presented work describes comparative studies of DLC and black chrome coated SUS304 plates for intracorporeal medical devices. The authors demonstrate a set of results from various techniques to evaluate the adhesion, friction characteristics, durability, etc.

The work is worthy of publication, however it requires a major revision prior to its acceptance. Moreover, this work in many aspects is similar to the published article K. Sakurai, et al., Diamond & Related Materials 96 (2019) 97-103. The authors has to clearly state a novelty of this work.

The following comments need to be addressed by the authors prior to acceptance:

  1. How was the film thickness confirmed? What method?
  2. Why the Black Cr/SUS thickness was higher than DLC/SUS?
  3. The sentence “Friction strongly depends on the surface condition and the usage environment of the material [4, 22-25]” (page 2-3, line 84-85) is very vague. Please clarify. Moreover, this exact sentence appears twice in the manuscript – line 122.
  4. The reference [27] appears prior to reference [26] in the manuscript body text. Please correct.
  5. Please describe a bit more (1-2 more sentences) regarding the oxygen plasma treatment.
  6. Table 1 is a repetition of Table 1 in previous work [26]: K. Sakurai, et al., Diamond & Related Materials 96 (2019) 97-103.
  7. The protocols described for measuring the coefficient of friction, a high-pressure steam and sterilization conditions are well-documented in work [26] – REPETATION!!!
  8. There is a spealing mistake in section 2.5 “Light reflection evaluation before and after cleaniNg”.
  9. The standard deviation of critical load is missing.
  10. Statistics are needed to section 3.2.
  11. Explain how do you analyze significant morphological changes by EDS? Normally, EDS is used to evaluate chemical/compositional changes.

Moreover, English grammar has to be corrected almost in the entire manuscript. Comprehensive scientific proofreading and editing by native speakers of English need to be conducted.

Round 2

Reviewer 3 Report

The suggested comments were properly considered by the authors.